# Health-related quality of life and treatment satisfaction in Chronic Lymphocytic Leukemia (CLL) patients on ibrutinib compared to other CLL treatments in a real-world US cross sectional study

Kathleen L. Deering[1][◯]*, Murali Sundaram[2][◯], Qing Harshaw[1][◯], Jeremiah Trudeau[3][‡], Jacqueline Claudia Barrientos[4][‡]

1 EPI-Q, Inc., Oak Brook, Illinois, United States of America, 2 Janssen Scientific Affairs, LLC, Horsham, Pennsylvania, United States of America, 3 Janssen Global Services, LLC, Horsham, Pennsylvania, United States of America, 4 Mount Sinai Comprehensive Cancer Center, Miami Beach, Florida, United States of America

◯ These authors contributed equally to this work.
‡ JT and JCB also contributed equally to this work.
* katie.deering@epi-q.com

**Data Availability Statement:** Data can be found at https://data.mendeley.com/datasets/stz6fjfvn2/1.

## Abstract

The objective of this study was to describe real-world health-related quality of life (HRQoL) and treatment satisfaction of ibrutinib-treated patients with CLL compared to a reference group. This study was completed in two parts. The first portion (Norming Study) was a US online survey conducted to serve as a reference population. The Norming Study included a total of 139 patients with CLL, excluding those treated with ibrutinib: 64 were treatment naive (Tx naive), 36 were 1st line (1L), and 38 were in or had completed ≥2 lines (2L+) patients with CLL. The second portion (CLL Ibrutinib Study) included 1L and 2L+ ibrutinib patients with CLL treated for ≥6 months in which 118 patients (1L n = 88 and 2L+ n = 30) completed the study. Respondents completed demographic and clinical information and the following HRQoL surveys: (Short Form-12v2® Health Survey [SF-12v2], Functional Assessment of Cancer Therapy-General [FACT-G], FACT-Leukemia [FACT-Leu] Functional Assessment of Chronic Illness Therapy [FACIT]-Fatigue, and Cancer Therapy Satisfaction Questionnaire [CTSQ]). Higher scores indicate better HRQoL/treatment satisfaction. Differences in effect sizes between the two samples at the group level were calculated using Hedges' g. Medium to large positive effects were seen in the CLL Ibrutinib group on several measures compared to the Reference Study groups. The FACT-G total score was 89.2 ±11.1 for CLL Ibrutinib Study patients compared to 75.8±22.6 CLL Norming Tx naïve patients, 61.3±21.8 in 1L, and 61.7±20.7 in 2L+. Similar trends were seen with FACT-Leu total score and FACIT-Fatigue. CLL Ibrutinib Study patients scored higher on all CTSQ domain scores compared to the CLL Norming patients treated with other CLL therapies. We found that Ibrutinib-treatment had better HRQoL and treatment satisfaction compared to patients receiving other therapies, irrespective of line of therapy.

Deering, Katie (2022), "Cross-Sectional Patient-Reported Outcomes in Chronic Lymphocytic Leukemia Patients ", Mendeley Data, V1, doi: 10. 17632/stz6fjfvn2.1.

**Funding:** This study was sponsored by Janssen Scientific Affairs, LLC (Janssen). MS was an employee of Janssen, and JT is currently an employee of Janssen. KLD and QH are employees of EPI-Q Inc., which received payment from Janssen Scientific Affairs associated with the development and execution of this study. All authors contributed to the study design and interpretation of the data.

**Competing interests:** MS was an employee of Janssen Scientific Affairs, LLC (Janssen), and JT is currently an employee of Janssen. KLD and QH are employees of EPI-Q Inc., which received payment from Janssen associated with the development and execution of this study. JCB has received honoraria for consulting for Gilead, AstraZeneca, Janssen, Sandoz, Genentech, Bayer, and Celgene. JCB has received honoraria for consulting for Gilead, AstraZeneca, Janssen, Sandoz, Genentech, Bayer, Beigene, Pharmacyclics/AbbVie, and Celgene/BMS. JCB has received research support from Gilead, AstraZeneca, and Pharmacyclics/AbbVie. This does not alter our adherence to PLOS ONE policies on sharing data and materials. Grant Recipient: EPI-Q, Inc.

# Introduction

Chronic lymphocytic leukemia (CLL) is the most common form of leukemia in the United States (US) and Western Europe, with approximately 20,000–21,000 newly diagnosed patients with CLL in the US every year [1, 2]. This disease most commonly presents in an elderly population and males are more likely to be afflicted with CLL than females (1.6:1) [1, 2].

The clinical presentation of CLL is highly variable. Many patients will be asymptomatic and diagnosed incidentally, while some patients are symptomatic at the time of diagnosis requiring immediate therapy. Upon diagnosis of CLL, the initial step is to determine whether the patient requires treatment based on significant disease-related symptoms or is a candidate for observation [3].

Despite recent advances in the treatment of CLL, relapse is common. The condition substantially impacts health-related quality of life (HRQoL) [4–7]. HRQoL is often not assessed in real-world studies; hence, there is limited understanding about the real-world HRQoL of patients diagnosed with CLL, especially as it relates to the patients' therapeutic disposition (e.g., active surveillance, 1$^{st}$ line [1L] treatment, and relapse or $\geq$2 line [2L+] treatments) [5–7]. Studies evaluating HRQoL have largely focused on comparing treated and untreated populations [4, 8–10]. Studies comparing treatment regimens have only seen small differences; however, the heterogeneity in symptomatic presentation may tend to underestimate such differences as studies focusing on the more symptomatic subgroup of patients have found greater treatment impacts [6, 7, 11].

More data on the impact of treatment on HRQoL is important to understand the overall impact of the therapy on the patients' well-being given the chronic nature of CLL. While incorporating HRQoL into clinical trials is important, the generalizability of these results may be limited due to the inclusion and exclusion criteria. The objective of this study was to describe HRQoL and treatment satisfaction of ibrutinib-treated patients with CLL outside of a clinical trial context with commercially available ibrutinib use and compare it with patients who are treatment naive or receiving another CLL-directed therapy (excluding ibrutinib).

# Methods

This study was conducted in two parts. The first part of the study (CLL Norming) was conducted between November 2019 and February 2020 and consisted of surveying patients with CLL from a US-based panel of patients who previously registered with Nielsen Opinion Quest. The objective of the first study was to generate reference data for HRQoL and treatment satisfaction in a real-world population of patients with CLL, excluding those treated with ibrutinib. This study also included patients with (1) other cancers and a (2) general population without cancer as reference populations; however, this paper focuses solely on the results of the CLL population.

The CLL Norming Study was a cross-section of patients reporting they were diagnosed with CLL and receiving treatment other than ibrutinib or those that were treatment naive (i.e., patients on active surveillance). The pool of potential respondents with CLL received an invitation email to participate in this study. Those who proceeded to the survey were presented with an electronic informed consent at the onset which was documented in the electronic database and approved by the IRB. Written consent was not feasible as patients were contacted via email to participate in the online survey. Patients who were eligible and willing to provide informed consent were enrolled and divided into 1L, 2L+, and treatment naive groups.

The second study (CLL Ibrutinib Study) was a survey of US patients with CLL treated with ibrutinib recruited from either their treating physician or specialty pharmacies. For the CLL Ibrutinib Study, specialty pharmacies recruited qualifying patients with CLL between January

2019 and April 2019 and oncologists/hematologists recruited and identified qualifying patients with CLL between December 2019 and February 2020. Specialty pharmacies and oncologists/ hematologists were selected on a convenience basis from national databases. Clinicians/sites only accessed patient medical records to confirm the patient met the inclusion criteria and were eligible to be recruited for the study. Patients were eligible if they were 18 years of age or older, had a diagnosis of CLL, and been on ibrutinib for CLL for a minimum of 6 months regardless of line of therapy to understand the tolerability and HRQOL with extended treatment. Those who were willing to proceeded with the survey were presented with a verbal or electronic informed consent and documented in a secure electronic database and approved by the IRB. Written consent was not feasible as patients were contacted via phone or email to participate in the phone/online survey. Patients eligible and willing to provide informed consent were enrolled and divided into 1L and 2L+ groups.

The study was approved by the WCG–New England Institutional Review Board (#120190385). No formal sample size calculation was performed for these studies. Surveys were administered via the internet or the phone and took approximately 25–30 minutes to complete and participants were compensated for their time. The recruiting partners did not provide the survey completion rates. The data collected included: demographic, clinical information, and HRQOL and treatment satisfaction patient-reported outcome (PRO) questionnaires.

## Patient-reported outcome instruments

**HRQoL instruments.** *Short Form-12v2® Health Survey (SF-12v2)*. The SF-12v2 is a 12-item survey for measuring general functional health and well-being including items such as: physical functioning, role limitations due to physical problems, bodily pain, general health, vitality, social functioning, role of limitations due to emotional problems, and mental health. The Physical Component Summary and the Mental Component Summary are the two summary scores generated from the questionnaire. Scores for each of the summaries range from 0 to 100, where 0 indicates the lowest level of health and 100 indicates the highest level of health, and the national norm mean score is 50 with a standard deviation of 10 [12].

*Functional Assessment of Cancer Therapy-General (FACT-G)*. The FACT-G is a general health scale that consists of a 27-item self-report questionnaire. The FACT-G includes the following domains/subscales:

- Physical Well-Being (PWB) subscale includes items related to fatigue, nausea, illness, pain, etc.

- Social Well-Being (SWB) subscale tends not to change much in a clinical trial of a drug therapy for a medical condition.

- Emotional Well-Being (EWB) subscale includes items related to sadness, nervousness, and worry about their disease and outcomes.

- Functional Well-Being (FWB) subscale includes items related to productivity at work and in daily activities/living.

The FACT-G score ranges from 0 to 108 and higher scores represent higher level of health [13].

*Functional Assessment of Cancer Therapy-Leukemia (FACT-Leu)*. FACT-Leu is a disease-specific scale that consists of a 44-item self-report questionnaire. It includes the 27-item FACT-general (FACT-G, above) and a 17-item subscale that assesses symptoms and concerns related to leukemia (FACT-LeuS). The FACT-Leukemia total score is derived from the

FACT-G and FACT-LeuS scores. The FACT-Leu total score ranges from 0 to 176 where higher scores represent higher level of health [14].

*Functional Assessment of Chronic Illness Therapy (FACIT)–Fatigue.* The FACIT-Fatigue is a 13-item scale that assesses self-reported fatigue and its impact on daily activities and function. The total FACIT-Fatigue score ranges from 0 to 52, where higher scores represent less fatigue [15, 16].

**Treatment satisfaction instrument.** *Cancer Therapy Satisfaction Questionnaire (CTSQ).* The CTSQ measures treatment satisfaction in individuals with cancer based on three domains: expectations of therapy (ET), feelings about side effects (FSE), and satisfaction with therapy (SWT). Domain scores range from 0 to 100, with a higher score representing a better outcome [17, 18].

## Analysis methods

Descriptive analyses were used to understand patient demographics, comorbidities, and PROs and summarized using proportions, means and standard deviations, where appropriate. The results of validated instruments were scored according to the appropriate algorithm.

Effect sizes of the difference in HRQoL and treatment satisfaction scores at the group level between the CLL Norming and CLL Ibrutinib studies were calculated using Hedges' g, which provides a measure of effect size weighted according to the relative size of each sample, given the different sample sizes among the groups [19]. An effect size of 0.2 is considered a small effect, 0.5 is a medium effect, and 0.8 is a large effect and the sign of the effect indicates the direction of the effect [20].

## Results

### Patient and site characteristics

All patients that qualified and started the surveys did were able to complete them regardless of mechanism (online or phone) and there was no missing data. The CLL Norming Study included a total of 138 patients with CLL (n = 64 treatment naive [Tx naive], n = 36 1st line [1L], and n = 38 2+ lines [2L+]). The CLL Ibrutinib Study included a total of 118 patients with CLL treated with ibrutinib (n = 88 in 1L and n = 30 in 2L+). Of the 118 patients with CLL treated with ibrutinib, 23 patients were recruited from specialty pharmacies (n = 11 in 1L and n = 12 in 2L+). All patients that qualified and started the survey completed the survey regardless of the route used (online or phone).

The mean age of participants was 66.7±8.6 years for the CLL Ibrutinib Study patients and the majority were male (66.1%), Caucasian (79.7%), married (72.9%) and had some college education (78.0%). Comparatively, the CLL Norming Study population was younger (58.3 ±14.4 years) with nearly equal numbers of male and females (45.7% and 54.3%, respectively). These trends are illustrated in Table 1 and are similar for the treatment line groups among the CLL Ibrutinib Study and CLL Norming Study patients.

Overall, the CLL Ibrutinib Study patients had higher rates for most comorbidities versus the CLL Norming Study patients with the exception of rheumatologic disease, mental health, and heart failure. Among the CLL Norming Study, the comorbidity rates were generally higher in the Tx Naive group followed by the 1L group whereas in the CLL Ibrutinib Study patients the rates were higher in the 2L+ group (Table 1).

Majority of patients in the 2L+ groups of the CLL Norming and Ibrutinib Studies were on their 2L treatment (83.3% for CLL Ibrutinib Study and 65.8% for CLL Norming Study). Of the patients in the CLL Norming Study 1L group, the most common reported treatments were fludarabine, cyclophosphamide, and rituximab (FCR, 41.7%), rituximab (27.8%), and fludarabine

**Table 1. Respondent characteristics.**

| | CLL Norming Tx Naïve N = 64 | CLL Norming 1st Line N = 36 | CLL Norming 2 + Lines N = 38 | CLL Ibrutinib Total N = 118 | CLL Ibrutinib 1st Line N = 88 | CLL Ibrutinib 2 + Lines N = 30 |
|---|---|---|---|---|---|---|
| Age, mean ±SD | 61.6 ±12.2 | 56.8 ±14.2 | 54.3 ±16.9 | 66.7 ±8.6 | 65.9 ±9.1 | 69.0 ±6.4 |
| Gender, n(%) | | | | | | |
| Male | 30 (46.9) | 15 (41.7) | 18 (47.4) | 76 (66.4) | 56 (63.6) | 20 (66.7) |
| Female | 34 (53.1) | 21 (58.3) | 20 (52.6) | 42 (35.6) | 32 (36.4) | 10 (33.3) |
| Race, n(%) | | | | | | |
| Caucasian/White | 59 (92.2) | 28 (77.8) | 31 (81.6) | 94 (79.7) | 68 (77.3) | 26 (86.7) |
| African-American/ Black | 3 (4.7) | 4 (11.1) | 6 (15.8) | 18 (15.3) | 15 (17.0) | 3 (10.0) |
| Asian | 2 (3.1) | 1 (2.8) | 0 | 5 (4.2) | 5 (5.7) | 0 |
| Other | 0 | 3 (8.3) | 1 (2.6) | 0 | 0 | 0 |
| Prefer not to say | 0 | 0 | 0 | 1 (0.85) | 0 | 1 (3.3) |
| Ethnicity, n(%) | | | | | | |
| Not Hispanic/Latino | 60 (93.8) | 30 (83.3) | 31 (81.6) | 109 (92.4) | 85 (96.6) | 24 (80.0) |
| Hispanic/Latino | 4 (6.3) | 4 (11.1) | 7 (18.4) | 8 (6.8) | 3 (3.4) | 5 (16.7) |
| Prefer not to say | 0 | 2 (5.6) | 0 | 1 (0.85) | 0 | 1 (3.3) |
| Marital Status, n(%) | | | | | | |
| Married | 49 (76.6) | 22 (61.1) | 25 (65.8) | 86 (72.9) | 69 (58.5) | 17 (56.7) |
| Separated | 3 (4.7) | 1 (2.8) | 1 (2.6) | 6 (5.1) | 3 (3.4) | 3 (10.0) |
| Divorced | 7 (10.9) | 4 (11.1) | 2 (5.3) | 7 (5.9) | 5 (5.7) | 2 (6.7) |
| Widowed | 2 (3.1) | 4 (11.1) | 3 (7.9) | 13 (11.0) | 8 (9.1) | 5 (16.7) |
| Single, never married | 3 (4.7) | 5 (13.9) | 7 (18.4) | 2 (1.7) | 1 (1.1) | 1 (3.3) |
| Prefer not to say | 0 | 0 | 0 | 4 (3.4) | 2 (2.3) | 2 (6.7) |
| Level of Education, n (%) | | | | | | |
| Elementary (Grade 1–8) | 0 | 0 | 0 | 0 | 0 | 0 |
| High school (or GED) | 15 (23.4) | 7 (19.4) | 5 (13.2) | 2 (1.7) | 2 (2.3) | 0 |
| Junior college | 12 (18.8) | 6 (16.7) | 8 (21.1) | 23 (19.5) | 18 (20.5) | 5 (16.7) |
| College | 17 (26.6) | 15 (41.7) | 18 (47.4) | 11 (9.3) | 6 (6.8) | 5 (16.7) |
| Post-graduate | 20 (31.3) | 8 (22.2) | 7 (18.4) | 55 (46.6) | 43 (48.9) | 12 (40.0) |
| Prefer not to say | 0 | 0 | 0 | 27 (22.9) | 19 (21.6) | 8 (26.7) |
| Comorbidities, n(%) | | | | | | |
| Rheumatologic Disease | 6 (9.4) | 6 (16.7) | 5 (13.2) | 6 (5.1) | 3 (3.4) | 3 (10.0) |
| High Cholesterol | 30 (46.9) | 12 (33.3) | 8 (21.1) | 59 (50.0) | 36 (40.9) | 23 (76.7) |
| Diabetes | 11 (17.2) | 9 (25.0) | 3 (7.9) | 22 (18.6) | 15 (17.0) | 7 (23.3) |
| Heart Disease | 6 (9.4) | 3 (8.3) | 0 | 18 (15.3) | 14 (15.9) | 4 (13.3) |
| Lung Disease | 4 (6.3) | 0 | 0 | 14 (11.9) | 10 (11.4) | 4 (13.3) |
| Heart Failure | 1 (1.6) | 1 (2.8) | 0 | 2 (1.7) | 2 (2.3) | 0 |
| Peripheral Vascular Disease | 1 (1.6) | 1 (2.8) | 0 | 5 (4.2) | 5 (5.7) | 0 |
| Myocardial Infarction | 3 (4.7) | 1 (2.8) | 0 | 11 (9.3) | 7 (8.0) | 4 (13.3) |
| Gastrointestinal Disease | 7 (10.9) | 4 (11.1) | 2 (5.3) | 12 (10.2) | 10 (11.4) | 2 (6.7) |
| Mental Health Issues | 9 (14.1) | 6 (16.7) | 1 (2.6) | 8 (6.8) | 4 (4.5) | 4 (13.3) |
| High Blood Pressure | 31 (48.4) | 12 (33.3) | 6 (15.8) | 60 (50.8) | 41 (46.6) | 19 (63.3) |
| Arthritis | 11 (17.2) | 5 (13.9) | 4 (10.5) | 24 (20.3) | 20 (22.7) | 4 (13.3) |
| Kidney Disease | 4 (6.3) | 1 (2.8) | 1 (2.6) | 7 (5.9) | 3 (3.4) | 4 (13.3) |

*(Continued)*

**Table 1.** (Continued)

|  | CLL Norming Tx Naïve N = 64 | CLL Norming 1st Line N = 36 | CLL Norming 2 + Lines N = 38 | CLL Ibrutinib Total N = 118 | CLL Ibrutinib 1st Line N = 88 | CLL Ibrutinib 2 + Lines N = 30 |
|---|---|---|---|---|---|---|
| Comorbidities, mean ±SD | 2.0 ±1.7 | 0.9 ±1.4 | 1.5 ±1.8 | 2.1 ±1.6 | 1.9 ±1.5 | 2.6 ±1.9 |
| 3+ Comorbidities | 25 (39.1) | 7 (19.4) | 9 (23.7) | 46 (39.0) | 30 (34.1) | 16 (53.3) |

and rituximab FR, 13.9%). Rituximab (39.5%), FCR (23.7%), and bendamustine and rituximab (BR, 18.4%) were the most common reported treatments in the CLL Norming Study 2L + group. Only 5.6% reported not currently receiving treatment in the CLL Norming 1L, whereas 15.8% of the CLL Norming 2L+ reported they were not currently receiving treatment (Table 2). The time on or off chemotherapy was not evaluated as a significant portion of responses were missing or marked unknown.

In the CLL Ibrutinib 2L+ group, only 3 patients reported their previous treatments; the remaining was missing or unknown. Approximately 50% of CLL Ibrutinib 1L patients reported being on ibrutinib between 6 months and 1 year and nearly 40% reported being on ibrutinib between 1 year and 2 years. The reported duration on ibrutinib for the CLL Ibrutinib

**Table 2. Respondent treatment characteristics.**

|  | CLL Norming No Tx N = 64 | CLL Norming 1st Line N = 36 | CLL Norming 2 + Lines N = 38 | CLL Ibrutinib Total N = 118 | CLL Ibrutinib 1st Line N = 88 | CLL Ibrutinib 2 + Lines N = 30 |
|---|---|---|---|---|---|---|
| Line of Treatment, n(%) |  |  |  |  |  |  |
| Patients no started treatment | 64 (100.0) | - - - - | - - - - | - - - - | - - - - | - - - - |
| Patients on initial treatment | - - - - | 36 (100.0) | - - - - | 88 (74.6) | 88 (100) | - - - - |
| Patients on second-line treatment | - - - - | - - - - | 25 (65.8) | 25 (21.2) | - - - - | 25 (83.3) |
| Patients on third-line treatment | - - - - | - - - - | 5 (13.2) | 3 (2.5) | - - - - | 3 (10.0) |
| Patients on more than three lines of treatment | - - - - | - - - - | 8 (21.1) | 2 (1.7) | - - - - | 2 (6.7) |
| Current chemotherapy treatments, n (%) |  |  |  |  |  |  |
| Ibrutinib | - - - - | - - - - | - - - - | 118 (100) | 88 (100) | 30 (100) |
| Idelalisib | - - - - | 2 (5.6) |  | - - - - | - - - - | - - - - |
| Venetoclax | - - - - | 2 (5.6) | 4 (10.5) | - - - - | - - - - | - - - - |
| Rituximab | - - - - | 10 (27.8) | 15 (39.5) | - - - - | - - - - | - - - - |
| BR (bendamustine and rituximab) | - - - - | 2 (5.6) | 7 (18.4) | - - - - | - - - - | - - - - |
| FCR (fludarabine, cyclophosphamide, and rituximab) | - - - - | 15 (41.7) | 9 (23.7) | - - - - | - - - - | - - - - |
| FR (fludarabine and rituximab) | - - - - | 5 (13.9) | 2 (5.3) | - - - - | - - - - | - - - - |
| Obinutuzumab | - - - - | 3 (8.3) | 1 (2.6) | - - - - | - - - - | - - - - |
| Acalabrutinib | - - - - | 0 | 1 (2.6) | - - - - | - - - - | - - - - |
| Chlorambucil | - - - - | 0 | 1 (2.6) | - - - - | - - - - | - - - - |
| Currently not on treatment | - - - - | 2 (5.6) | 6 (15.8) | - - - - | - - - - | - - - - |
| Other | - - - - | 2 (5.6) | 3 (7.9) | - - - - | - - - - | - - - - |
| Approximately how long have you been on ibrutinib?, n(%) |  |  |  |  |  |  |
| 6 months and up to 1 year | - - - - | - - - - | - - - - | 58 (49.2) | 45 (51.1) | 13 (43.3) |
| More than 1 year and up to 2 years | - - - - | - - - - | - - - - | 45 (38.1) | 35 (39.8) | 10 (33.3) |
| More than 2 years | - - - - | - - - - | - - - - | 15 (12.7) | 8 (9.1) | 7 (23.3) |

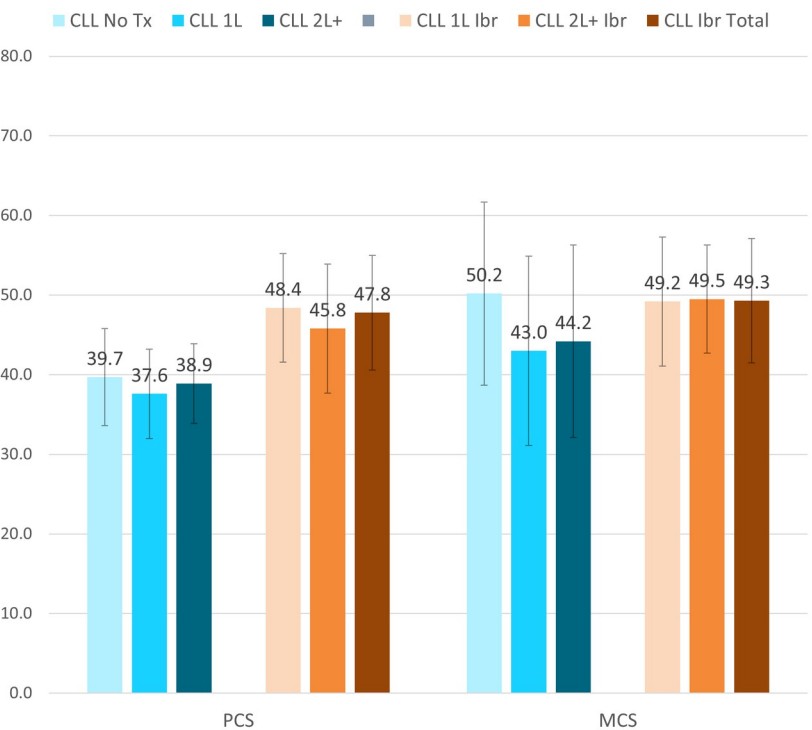

**Fig 1. Short Form-12v2® Health Survey (SF-12v2) at the group level, mean ± SD.** CLL = Chronic Lymphocytic Leukemia; Ibr = Ibrutinib; MCS = Mental Component Summary; PCS = Physical Component Summary; Tx = Treatment. PCS score range: 0–80; MCS score range: 0–80. Higher scores representing better outcomes. All comparisons of the CLL Ibrutinib groups to the CLL Norming Study 1L and 2L+ groups had p-vales <0.05.

2L+ was 43.3% between 6 months and 1 year, 33.3% between 1–2 years and 23.3% more than 2 years (Table 2).

## HRQoL and treatment satisfaction

For the SF-12v2, the CLL Ibrutinib Study patients scored higher indicating better health on the Physical Component Summary dimension (Total 47.8±7.2, 1L 48.4±6.8, and 2L+ 45.8±8.1) contrasted to the CLL Norming Study groups (Tx Naive 39.7±6.1, 1L 37.6±5.6, 2L+ 38.9±5.0). The CLL Ibrutinib Study patients also had better health on the Mental Component Summary (Total 49.3±7.8, 1L 49.2±8.1, and 2L+ 49.5±6.8) versus to CLL Norming 1L and 2L+ patients (43.0±11.9 and 44.2±12.1, respectively) and the scores were similar to the CLL Norming Tx Naive patients (50.2± 11.5) (Fig 1).

The FACT-G total scores illustrated CLL Ibrutinib Study patients (Total 79.3±17.3, 1L 78.8 ±18.4, and 2L+ 81.2±14.0) had the greatest quality of life in contrast to the CLL Norming Study Tx Naive, 1L, and 2L+ patients (75.8±22.6, 61.3±21.8, and 61.7±21.8, respectively) (Fig 2). The total, 1L, and 2L+ CLL Ibrutinib Study patients also had better HRQoL based on the FACT-Leukemia total score (130.8±24.6, 130.5±26.3, and 131.9±19.2, respectively) compared patients to the CLL Norming Tx Naive (121.8±35.9), 1L (101.4±36.3), and 2L+ (99.1 ±33.3) (Fig 3). The subscales with the greatest differences were the leukemia-specific concerns subscale (LeuS), Functional Well-Being (FWB), and Physical Well-Being (PWB) (Fig 4).

Among the CLL groups, patients treated with ibrutinib had less fatigue, indicated by higher scores on FACIT-Fatigue (Total 38.7±8.0, 1L 39.2±8.3, and 2L+ 37.7±8.1, respectively)

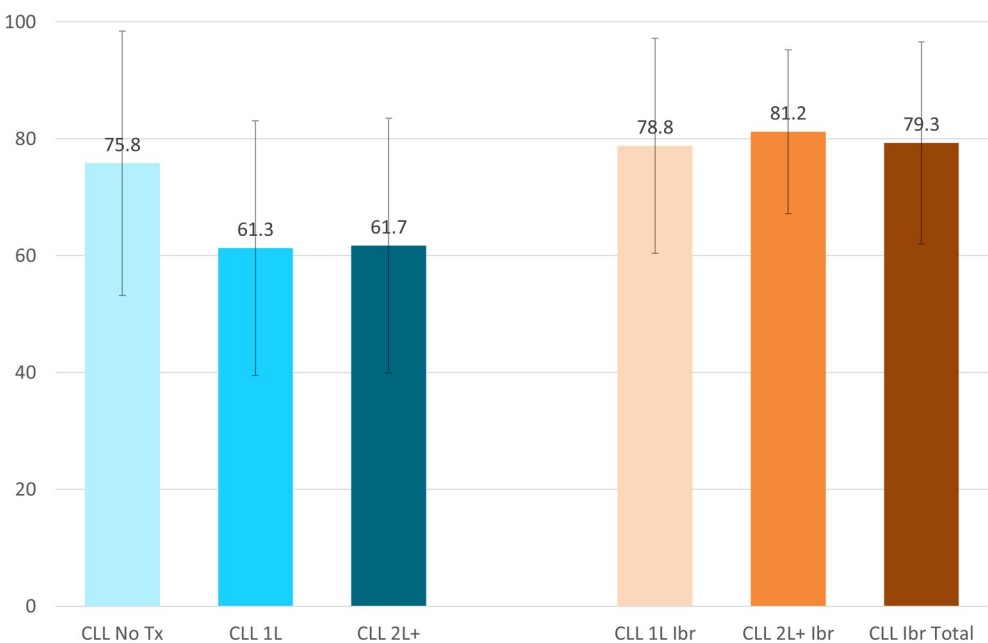

**Fig 2. Functional Assessment of Cancer Therapy-General (FACT-G) total score at the group level, mean ± SD.**
CLL = Chronic Lymphocytic Leukemia; Ibr = Ibrutinib; Tx = Treatment. FACT-G total score range: 0–108. Higher scores representing better outcomes. All comparisons of the CLL Ibrutinib groups to the CLL Norming Study 1L and 2L+ groups had p-vales <0.05.

compared to the CLL Norming Study Tx Naive (35.5±14.4), 1L (29.5±12.9), and 2L+ patients (28.7±13.5) (Fig 5).

Treatment satisfaction, as measured by the CTSQ, demonstrated that ibrutinib treated patients with CLL reported better scores on all three domains: expectations of therapy, feelings

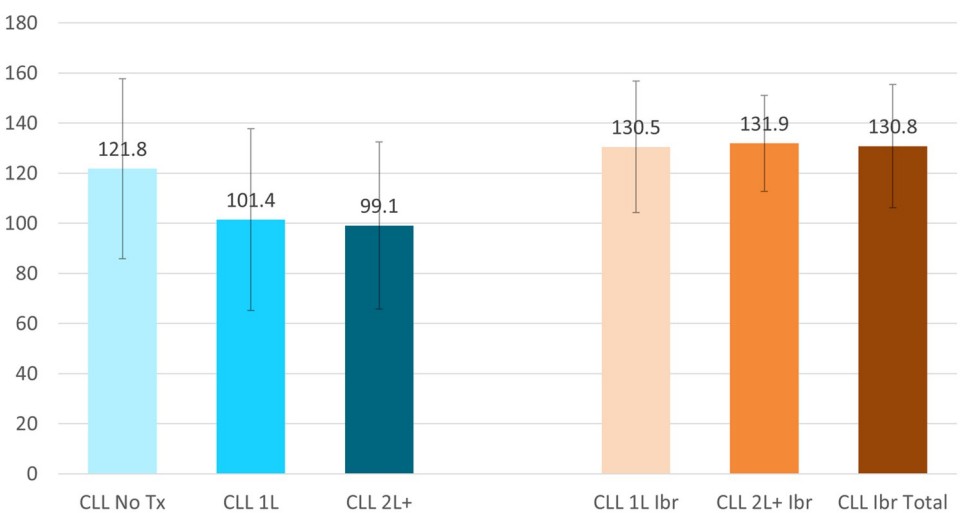

**Fig 3. Functional Assessment of Cancer Therapy-Leukemia (FACT-Leu) total score at the group level, mean ± SD.**
CLL = Chronic Lymphocytic Leukemia; Ibr = Ibrutinib; Tx = Treatment. FACT-Leu total score range: 0–176. Higher scores representing better outcomes. All comparisons of the CLL Ibrutinib groups to the CLL Norming Study 1L and 2L+ groups had p-vales <0.05.

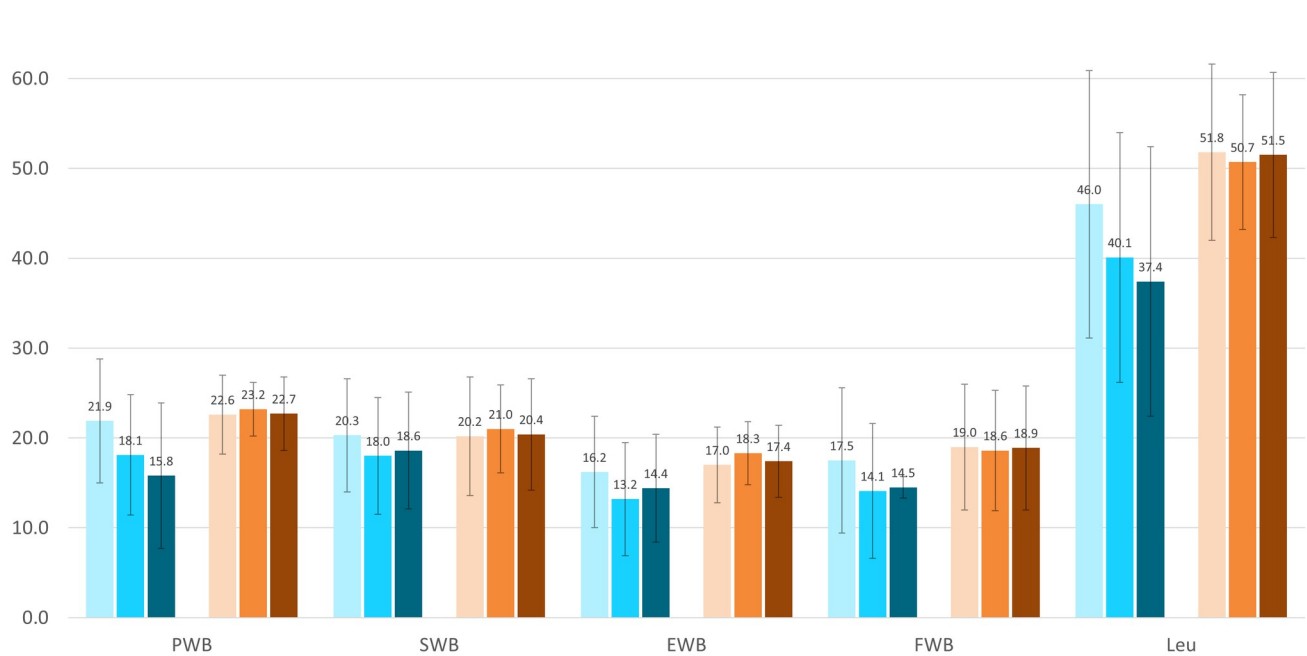

**Fig 4. Functional Assessment of Cancer Therapy-Leukemia (FACT-Leu) subscales at the group level, mean ± SD.** CLL = Chronic Lymphocytic Leukemia; EWB = Emotional Well-Being; FWB = Functional Well-Being; Ibr = Ibrutinib; Leu = Leukemia Subscale; PWB = Physical Well-Being; SWB = Social/Family Well-Being; Tx = Treatment. PWB score range: 0–28; SWB score range: 0–28; EWB score range: 0–24; FWB score range: 0–28; Leukemia Subscale score range: 0–68. Higher scores representing better outcomes. All comparisons of the CLL Ibrutinib groups to the CLL Norming Study 1L and 2L+ groups had p-vales <0.05.

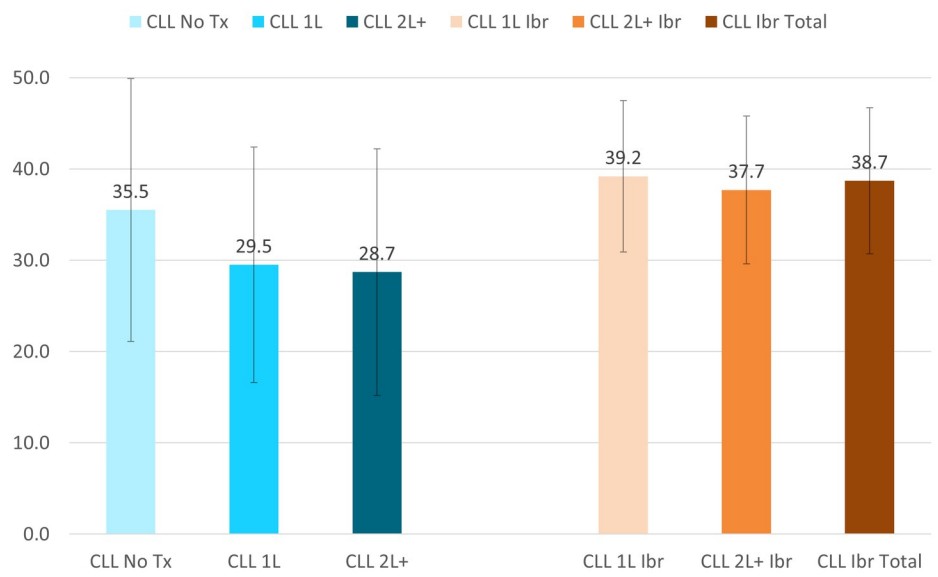

**Fig 5. Functional Assessment of Chronic Illness Therapy (FACIT)—Fatigue at the group level, mean ± SD.**
CLL = Chronic Lymphocytic Leukemia; Ibr = Ibrutinib; Tx = Treatment. FACIT-Fatigue score range: 0–52. Higher scores representing better outcomes. All comparisons of the CLL Ibrutinib groups to the CLL Norming Study 1L and 2L+ groups had p-vales <0.05.

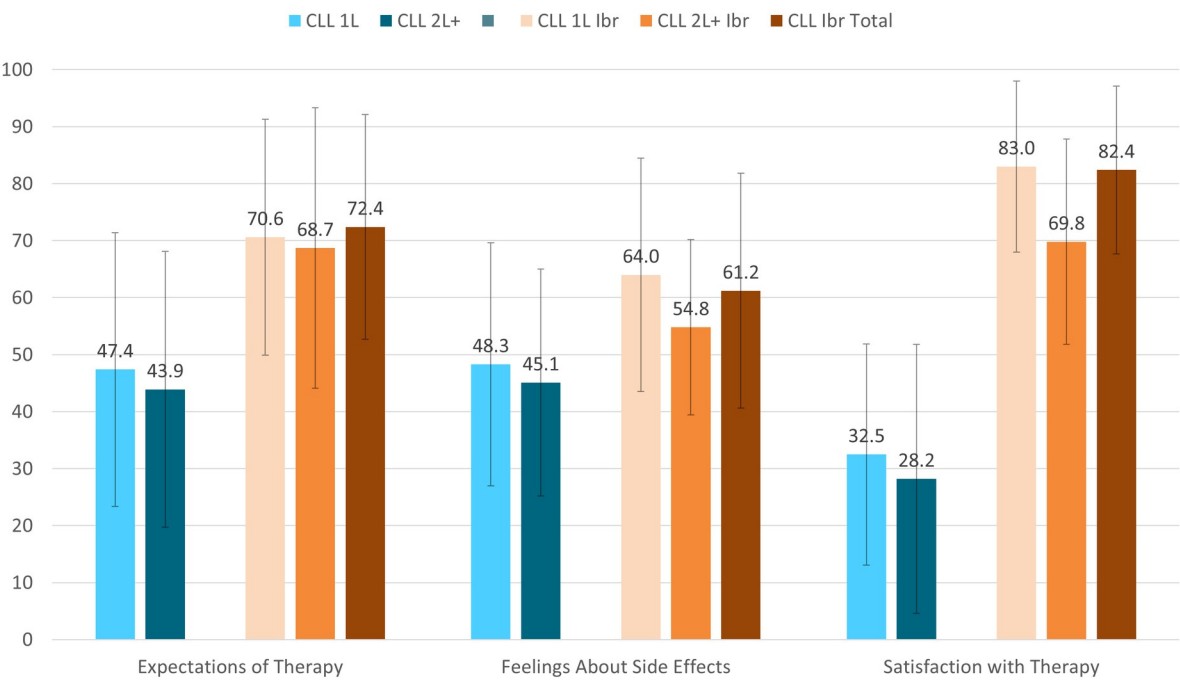

**Fig 6. Cancer Therapy Satisfaction Questionnaire (CTSQ) at the group level, mean ± SD.** CLL = Chronic Lymphocytic Leukemia; Ibr = Ibrutinib; Tx = Treatment;. Expectations of Therapy score range: 0–100; Feelings about Side Effects score range: 0–100; Satisfaction with Therapy score range: 0–100. Higher scores representing better outcomes. All comparisons of the CLL Ibrutinib groups to the CLL Norming Study 1L and 2L+ groups had p-vales <0.05.

about side effects, and satisfaction with therapy compared to the patients with CLL receiving other treatments (Fig 6).

Evaluating the differences in effect sizes determined the 1L and 2L+ ibrutinib treatment had generally large positive effects compared to patients with CLL treated with other therapies for 1L and 2L+ respectively (Fig 7). The two scales with medium effect size were the SF12v2 MCS for both the 1L and 2L+ comparisons (0.66 [95%CI: 026, 1.06] and 0.52 [95%CI: 0.03, 1.00], respectively) and the CTSQ FES for the 2L+ comparisons (0.53 [95%CI: 0.35, 1.02]).

Similarly, the effect sizes of ibrutinib treatment, regardless of treatment line, demonstrated positive effects when compared to the other CLL treatment 1L or 2L+ groups. The effect sizes of ibrutinib treatment compared to the other CLL treatments for 1L and 2L+ patients were: 1.7 [95%CI: 1.2, 2.1] and 1.5 [95%CI: 1.1, 1.9], respectively for the SF12v2 PCS dimension; 0.7 [95%CI: 0.3, 1.1] and 0.5 [95%CI: 0.2, 0.9], respectively for the SF12v2 MCS dimension; 0.9 [95%CI: 0.5, 1.3] and 1.0 [95%CI: 0.6, 1.3], respectively for the FACT-G; 1.1 [95%CI: 0.7, 1.4] and 1.2 [95%CI: 0.8, 1.6], respectively for the FACT-Leukemia; 1.0 [95%CI: 0.6, 1.4] and 1.0 [95%CI: 0.7, 1.4], respectively for the FACIT-Fatigue; 1.2 [95%CI: 0.8, 1.6] and 1.364 [95%CI: 1.0, 1.8], respectively for the CSTQ ET; and 3.1 [95%CI:2.6, 3.6] and 3.1 [95%CI: 2.6, 3.6], respectively for the CSTQ SWT (Fig 8).

The effect sizes of ibrutinib treatment compared to being treatment naive were generally small to neutral effects (SF12v2 MCS -0.1 [95%CI: -0.4, 0.2], FACT-G 0.2 [95%CI: -0.1, 0.5], FACT-Leu 0.3 [95%CI: 0.0, 0.6], and FACIT-Fatigue 0.3 [95%CI: -0.0, 0.6]) with the exception of a large positive effect on the SF12v2 PCS dimension (1.3 [95%CI: 0.9, 1.7]). (Fig 8).

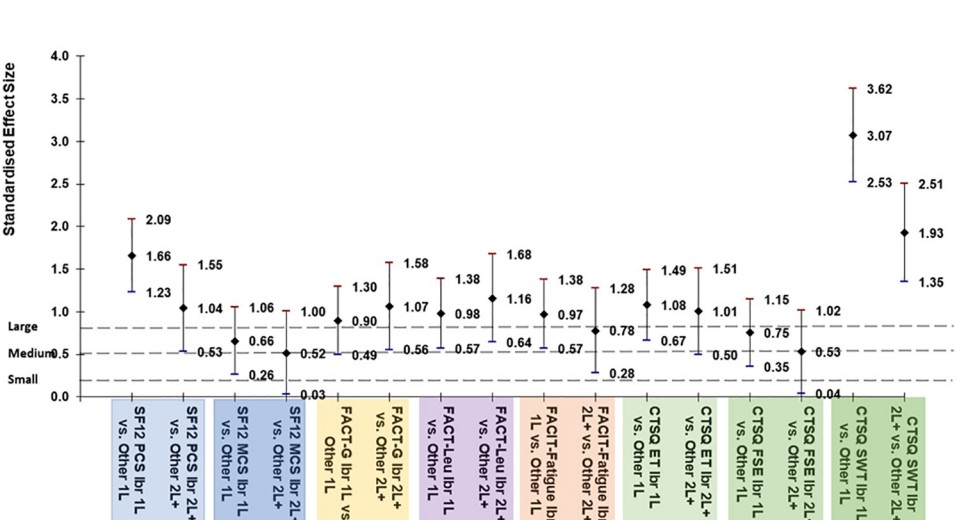

**Fig 7. Effect sizes of differences in HRQoL/treatment satisfaction scores at the group level.** CLL = Chronic Lymphocytic Leukemia; CTSQ = Cancer Therapy Satisfaction Questionnaire; FACIT = Functional Assessment of Chronic Illness Therapy; FACT = Functional Assessment of Cancer Therapy; Ibr = Ibrutinib; Leu = Leukemia; MCS = Mental Component Summary; PCS = Physical Component Summary; SF-12v2 = Short Form-12v2® Health Survey. An effect size of 0.2 is considered a small effect, 0.5 is a medium effect, and 0.8 is a large effect. The sign of the effect indicates the direction of the effect. [Cohen 1988].

## Discussion

Regardless of the line of therapy, in this cross-sectional CLL Ibrutinib Study patients reported better HRQoL and treatment satisfaction compared to the CLL Norming Study patients. This is demonstrated by the large positive effect sizes on generic and disease-specific HRQoL outcomes, feelings about side effects, and satisfaction with therapy of ibrutinib treatment in the CLL Ibrutinib Study versus other CLL treatments in the Norming Study.

Evaluating the HRQoL scores of the FACT-G relative to CLL published real-world literature and cancer norms, the physical, social/family, functional well-being, and overall HRQoL scores were similar to or better for the CLL Ibrutinib Study patients [21–23]. The CLL Ibrutinib Study patients also had similar HRQoL compared to the national population norms for the SF-12v2 (50±10) [24].

Similar to our study, a recently published study with patients in India evaluated HRQoL among active surveillance, CIT treated, and ibrutinib treated patients with CLL [25]. Youron et al found that patients on ibrutinib had better HRQoL compared with patients on CIT and had significantly higher social functioning score and reported statistically less fatigue and appetite loss compared to patients on CIT (p ≤ 0.05). Though not statistically different between the CLL groups, the patients on ibrutinib had the highest summary score followed by active surveillance. Additionally, the study demonstrated that CIT had worse global health (Odds Ratio 12.2) compared to active surveillance patients. Directly, these results are similar but is it important to note that the investigators used the European Organization for Research and Treatment of Cancer (EORTC) QLQ-C30 questionnaire and differences were evaluated based on small, medium, and large clinical significance guidelines for the EORTC QLQ-C30.

The primary goals of CLL treatment are to achieve and maintain remission for as long as possible to improve overall survival; particularly when on a drug that requires continuous

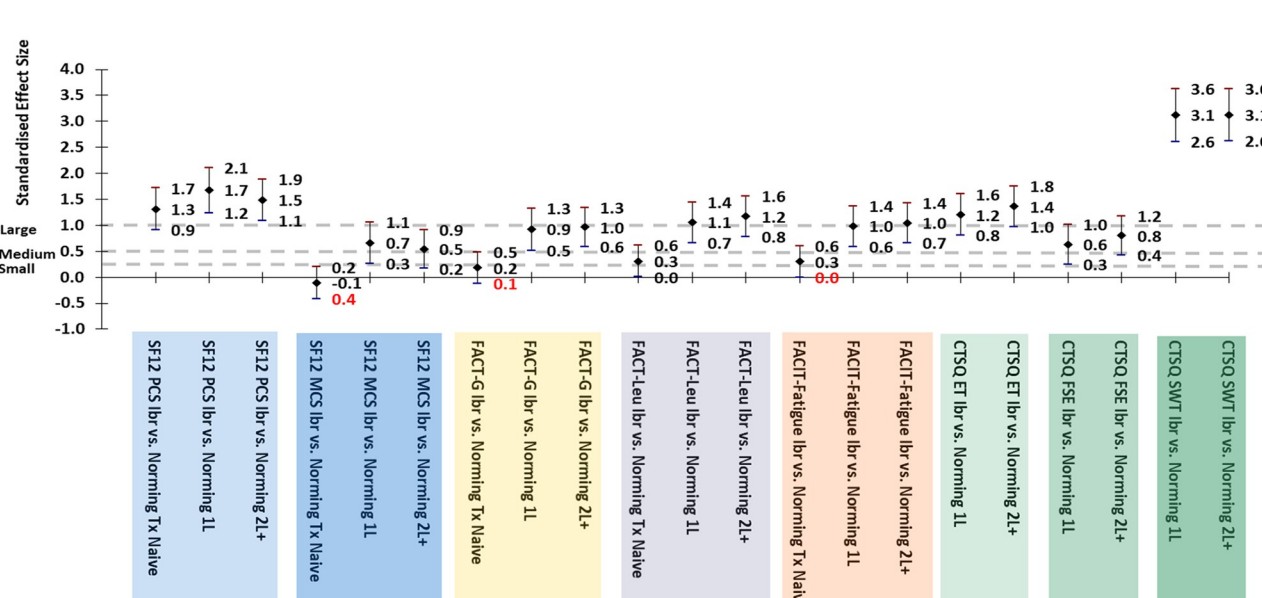

**Fig 8. Effect sizes of differences in HRQoL/treatment satisfaction scores for total Ibrutinib group compared to norming groups.** CLL = Chronic Lymphocytic Leukemia; CTSQ = Cancer Therapy Satisfaction Questionnaire; FACIT = Functional Assessment of Chronic Illness Therapy; FACT = Functional Assessment of Cancer Therapy; Ibr = Ibrutinib; Leu = Leukemia; MCS = Mental Component Summary; PCS = Physical Component Summary; SF-12v2 = Short Form-12v2® Health Survey. An effect size of 0.2 is considered a small effect, 0.5 is a medium effect, and 0.8 is a large effect. The sign of the effect indicates the direction of the effect. [Cohen 1988].

daily dosing. Treatment selection should include the impact of the disease and treatment HRQoL. Published data on HRQoL in CLL focusing on non-randomized controlled trials and real-world studies have previously found lower HRQoL in patients being treated for CLL (prior studies did not include ibrutinib-treated patients as this is a relatively new drug when compared to CIT) when compared to active surveillance patients [4, 8–10]. This is consistent with the current CLL Norming Study findings of lower HRQoL for patients on treatment. These results indicate CIT treatments despite their efficacy, may lead to detrimental impacts to HRQoL. This may be related to treatment failures, adverse events associated with treatment. Interestingly, the CLL Ibrutinib Study found higher HRQoL for patients treated with ibrutinib when compared to patients in active surveillance. It is possible that this is related to selection bias as patients on the ibrutinib cohort had been a minimum of 6 months, which may select for the patients that had already passed the critical time to develop toxicities. Another consideration is that since, per International Workshop on Chronic Lymphocytic Leukemia (iwCLL) guidelines, treatment is to be considered at the time that the patient is symptomatic, patients on active surveillance are actually feeling worse due to the disease itself [3]. Increasingly, HRQoL instruments are being incorporated into oncology clinical trials [26–29]. Two phase 3 studies of ibrutinib (RESONATE-2 and HELIOS) evaluated PRO measures. In the RESONATE-2 study comparing ibrutinib with ofatumumab in previously treated patients with CLL/SLL, ibrutinib demonstrated clinically meaningful improvement in fatigue and overall HRQoL based on the FACIT-Fatigue and EORTC QLQ-C30, respectively [30]. The HELIOS study evaluated the addition of continuous ibrutinib to up to six cycles of bendamustine plus

rituximab (BR) in previously treated patients with CLL/SLL [31]. For the HELIOS PRO analysis, the majority of patients had a moderate degree of fatigue and HRQoL impairment at baseline and the mean values were maintained over time [7]. However, a subgroup of patients with the worst fatigue and HRQoL at baseline showed greater improvements with ibrutinib plus BR. More research is needed to determine if these results were confounded by the concomitant treatment of ibrutinib and chemoimmunotherapy [11].

This study evaluates the impact of ibrutinib on HRQoL in patients treated with commercially available drugs outside of clinical trial participation to understand the impact of ibrutinib at different cross-sectional points in their treatment journey. Symptom and disease-specific characteristics have been shown to have important impact on the QOL of CLL patients [6, 7, 23]. Our study incorporated validated generic HRQoL, cancer-specific HRQoL, leukemia-specific HRQoL, and symptom-specific (fatigue) HRQoL instruments. Though this paper focuses on the CLL population, the Norming Study included groups of patients with other cancers (prostate, lung, multiple myeloma, other leukemias and lymphomas) and a general population group without cancer. Due to the inclusion of other cancer, the FACT-G was chosen as an instrument that could be administered to all cancer groups and the FACT-LeuS helped minimize the survey burden versus including the EORTC QLQ-C30 in addition to the FACT-G.

Clinical study populations that may be less symptomatic than a real-world population may tend to underestimate potential treatment impacts on HRQoL outside of a clinical trial setting (i.e., study patients may not have sufficient 'room for improvement' to show positive effects). Though the current study was cross-sectional in nature and did not assess pre-treatment or baseline scores for comparison, patients in the ibrutinib-treated cohort reported both higher expectations for treatment and higher satisfaction, suggesting that compared to other treatments, their expectations had been met.

The limitations of this study include potential issues with missing data, reliability, and recall bias in the data collection of clinical and treatment data since they were self-reported and not pathologically confirmed. Although it is known that *IGHV* and *TP53* mutation status affect response to CIT, which could affect HRQoL reports, it is also known that the rate of testing in routine clinical practice is less than optimal. In a recent study presented at ASH 2020, Mato and colleagues found that *TP53* mutation testing was performed in only 11% (n = 162) of pts and *IGHV* mutational status testing was performed in 12% Since all information was patient-reported and there was no option to confirm the reliability of the prognostic marker testing, the authors did not obtain this information given potential recall bias [32].

Another source of bias includes the fact that most respondents (87.4%) participated via an online survey which required participants to use a computer/tablet/phone and have internet access, hence limiting the opportunity of participation to patients with access to this technology. Selection biases may have occurred due to the different recruitment methods for each study and the exclusion of patients treated with ibrutinib for less than 6 months. Self-selection may have a risk of bias for the patients recruited through Nielsen. Importantly, these studies were a cross-sectional assessment of HRQoL including patients at all places in their treatment journey. As such, the varied sample sizes in each group and study, studies were not powered to detect differences, and results are not adjusted for previous treatments or duration of the reported current treatments.

In spite of the limitations of this study, we found the CLL Ibrutinib Study patients had better treatment satisfaction and HRQoL compared to the Norming Study Treatment Naive, 1L, and 2L+ patients on the cancer-, leukemia-, and fatigue-specific HRQoL instruments and when compared to patients treated with CIT. These results build upon evidence from clinical trials and serve to inform further investigations regarding the impact of ibrutinib on improving HRQoL in patients with a CLL diagnosis.

## Acknowledgments

The authors thank Emily Eastman PharmD, MBA, CSP and team at the University of Kentucky HealthCare—Specialty Pharmacy Services for their role in patient recruitment and survey administration. The authors would also like to thank Medical Data Analytics, a division of Market Certitude, L.L.C. an RTI Health Solutions Business for their support in site recruitment.

## Author Contributions

**Conceptualization:** Kathleen L. Deering, Murali Sundaram.

**Formal analysis:** Kathleen L. Deering, Qing Harshaw.

**Funding acquisition:** Murali Sundaram.

**Methodology:** Kathleen L. Deering, Murali Sundaram, Jeremiah Trudeau, Jacqueline Claudia Barrientos.

**Project administration:** Kathleen L. Deering.

**Supervision:** Kathleen L. Deering.

**Writing – original draft:** Kathleen L. Deering.

**Writing – review & editing:** Kathleen L. Deering, Murali Sundaram, Qing Harshaw, Jeremiah Trudeau, Jacqueline Claudia Barrientos.

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
