## [Decision Letter · Decision Letter 0]

9 Nov 2021

PONE-D-21-31586Health-related quality of life and treatment satisfaction in Chronic Lymphocytic Leukemia (CLL) patients on ibrutinib compared to other CLL treatments in a real-world US cross sectional studyPLOS ONE

Dear Dr. Deering,

Thank you for submitting your manuscript to PLOS ONE. After careful consideration, we feel that it has merit but does not fully meet PLOS ONE’s publication criteria as it currently stands. Therefore, we invite you to submit a revised version of the manuscript that addresses the points raised during the review process by Reviewer #2.

We look forward to receiving your revised manuscript.

Kind regards,

Francesco Bertolini, MD, PhD

Academic Editor

PLOS ONE

Journal Requirements:

a) Did participants provide their written or verbal informed consent to participate in this study? 

"MS was an employee of Janssen, and JT is currently an employee of Janssen. KLD and QH are employees of EPI-Q Inc., which received payment from Janssen

associated with the development and execution of this study. JCB has received

honoraria for consulting for Gilead, AstraZeneca, Janssen, Sandoz, Genentech, Bayer,

and Celgene. JCB has received research support from Gilead and

Pharmacyclics/AbbVie."

Reviewers' comments:

Reviewer's Responses to Questions

**Comments to the Author**

1. Is the manuscript technically sound, and do the data support the conclusions?

Reviewer #1: Yes

Reviewer #2: Partly

2. Has the statistical analysis been performed appropriately and rigorously? 

Reviewer #1: Yes

Reviewer #2: Yes

3. Have the authors made all data underlying the findings in their manuscript fully available?

Reviewer #1: No

Reviewer #2: Yes

4. Is the manuscript presented in an intelligible fashion and written in standard English?

Reviewer #1: Yes

Reviewer #2: Yes

5. Review Comments to the Author

Reviewer #1: The authors have addressed all concerns.

Reviewer #2: Here Deering et al report analyze PRO among patients with CLL treated with ibrutinib or not. The data suggest that ibrutinib treatment is associated with improved QOL, even as compared with patients who are treatment-naïve.

1. The authors do not present statistical analysis of their data, ie comparison between

2. Table 1 – it would be helpful to summarize comorbidities in this Table, in addition to listing individual diseases Perhaps, a Charlson comorbidity score could be used to achieve that. Was the severity of comorbidities known?

3. In connection with that, the Results show that patients in the Norming study had higher burden of comorbidities, despite being younger. Is there an explanation for that? Perhaps using an index like Charlson might help equalize the groups.

4. It is interesting that ibrutinib-treated patients reported improved QOL measures compared with treatment-naïve patients with CLL. The authors try to explain that by the requirement for therapy based on IWCLL criteria. However, it is presumed that the majority of CLL patients in the norming study are not symptomatic from CLL. However, they do have a higher burden of comorbidities. It is possible that comorbidities contribute to lower scores. This needs to be explained further/discussed as a limitation

5. Statistical significance should be presented on the graphs rather than be part of the legend. Indicate p<0.01 and p<0.05. Use of asterix is acceptable.

6. The authors should discuss what are clinically meaningful improvement in these scales?

7. There are multiple typos throughout the manuscript. Please check carefully. Also, please italicize gene names, such as IGHV and TP53.

6. PLOS authors have the option to publish the peer review history of their article (what does this mean?). If published, this will include your full peer review and any attached files.

Reviewer #1: **Yes: **Deepesh Lad

Reviewer #2: No

---

## [Author Response · Author response to Decision Letter 0]

11 May 2022

Journal Requirements:

RESPONSE: We have reviewed and made revisions as necessary.

a) Did participants provide their written or verbal informed consent to participate in this study? 

RESPONSE: We have added statements for why written consent was not obtained and noted that the procedure used was approved by the IRB.

"MS was an employee of Janssen, and JT is currently an employee of Janssen. KLD and QH are employees of EPI-Q Inc., which received payment from Janssen

associated with the development and execution of this study. JCB has received

honoraria for consulting for Gilead, AstraZeneca, Janssen, Sandoz, Genentech, Bayer,

and Celgene. JCB has received research support from Gilead and

Pharmacyclics/AbbVie."

RESPONSE: Please update the Competing Interests statement with the following “This does not alter our adherence to PLOS ONE policies on sharing data and materials.”

RESPONSE: We have uploaded a minimal anonymized data set and it can be found at .

Reviewer #1: 

The authors have addressed all concerns.

RESPONSE: Thank you for your thorough review and acceptance of our revisions. 

Reviewer #2: 

Here Deering et al report analyze PRO among patients with CLL treated with ibrutinib or not. The data suggest that ibrutinib treatment is associated with improved QOL, even as compared with patients who are treatment-naïve.

1. The authors do not present statistical analysis of their data, ie comparison between

RESPONSE: The study evaluated several populations and was not powered to detect differences among or between them. The authors did not feel it appropriate to generate statistical tests as that could lead to misinterpretation of the data.

2. Table 1 – it would be helpful to summarize comorbidities in this Table, in addition to listing individual diseases Perhaps, a Charlson comorbidity score could be used to achieve that. Was the severity of comorbidities known?

RESPONSE: Not all of the comorbidities in the Charlson comorbidity score were collected in the survey. The authors added the average number of comorbidities and the number and percentage of patients with 3+ comorbidities to try and better describe the burden. 

3. In connection with that, the Results show that patients in the Norming study had higher burden of comorbidities, despite being younger. Is there an explanation for that? Perhaps using an index like Charlson might help equalize the groups.

RESPONSE: As described in the results, the IBR population had a higher comorbidity burden, not the Norming study. Within the Norming study population, the Naïve patients were older and had a higher comorbidity burden. The comorbidities that were higher in the Norming Study Naïve population than the Norming Study 1L or 2L+ populations were generally diseases that increase with age. Additionally, the Norming Study Naïve population was twice the size as the other two Norming Study populations. Thus, the differences in comorbidities may be related to the age difference (~10 years) or the population size. We have included this in the limitations. 

4. It is interesting that ibrutinib-treated patients reported improved QOL measures compared with treatment-naïve patients with CLL. The authors try to explain that by the requirement for therapy based on IWCLL criteria. However, it is presumed that the majority of CLL patients in the norming study are not symptomatic from CLL. However, they do have a higher burden of comorbidities. It is possible that comorbidities contribute to lower scores. This needs to be explained further/discussed as a limitation

RESPONSE: This is a possibility and have included it in the limitations.

5. Statistical significance should be presented on the graphs rather than be part of the legend. Indicate p<0.01 and p<0.05. Use of asterix is acceptable.

RESPONSE: The authors feel that it is difficult to present the statistical significance on the graph. We felt an asterisk would not be able to distinguish that CLL Ibrutinib groups were not statistically significant compared to the CLL Norming Treatment Naïve group and brackets would make the graph very busy and hard to read. These p-values were requested by a past reviewer; however, the groups were not powered to detect differences and the authors do not want to over emphasize and risk misinterpretation of the data. If the reviewer has another suggestion, we are open to considering. 

6. The authors should discuss what are clinically meaningful improvement in these scales?

RESPONSE: Clinically meaningful improvements are generally reserved for measuring improvements for an individual from one time period to another. We only evaluated a cross-sectional measurement for groups and did not feel it was appropriate to evaluate and include clinically meaningful improvements.

7. There are multiple typos throughout the manuscript. Please check carefully. Also, please italicize gene names, such as IGHV and TP53. 

RESPONSE: Apologies for missing any typos following the last round of revisions. We have reviewed carefully and made corrections noted by the reviewer. Of note, some of the references used British English spellings in their titles so these are not corrected for American English.

---

## [Decision Letter · Decision Letter 1]

8 Jun 2022

Health-related quality of life and treatment satisfaction in Chronic Lymphocytic Leukemia (CLL) patients on ibrutinib compared to other CLL treatments in a real-world US cross sectional study

PONE-D-21-31586R1

Dear Dr. Deering,

We’re pleased to inform you that your manuscript has been judged scientifically suitable for publication and will be formally accepted for publication once it meets all outstanding technical requirements.

Kind regards,

Francesco Bertolini, MD, PhD

Academic Editor

PLOS ONE

Additional Editor Comments (optional):

Reviewers' comments:

Reviewer's Responses to Questions

**Comments to the Author**

1. If the authors have adequately addressed your comments raised in a previous round of review and you feel that this manuscript is now acceptable for publication, you may indicate that here to bypass the “Comments to the Author” section, enter your conflict of interest statement in the “Confidential to Editor” section, and submit your "Accept" recommendation.

Reviewer #1: All comments have been addressed

Reviewer #2: All comments have been addressed

2. Is the manuscript technically sound, and do the data support the conclusions?

Reviewer #1: Yes

Reviewer #2: Yes

3. Has the statistical analysis been performed appropriately and rigorously? 

Reviewer #1: Yes

Reviewer #2: Yes

4. Have the authors made all data underlying the findings in their manuscript fully available?

Reviewer #1: Yes

Reviewer #2: Yes

5. Is the manuscript presented in an intelligible fashion and written in standard English?

Reviewer #1: Yes

Reviewer #2: Yes

6. Review Comments to the Author

Reviewer #1: (No Response)

Reviewer #2: (No Response)

7. PLOS authors have the option to publish the peer review history of their article (what does this mean?). If published, this will include your full peer review and any attached files.

Reviewer #1: No

Reviewer #2: No
